# Sovateltide Mediated Endothelin B Receptors Agonism and Curbing Neurological Disorders

**DOI:** 10.3390/ijms23063146

**Published:** 2022-03-15

**Authors:** Amaresh K. Ranjan, Anil Gulati

**Affiliations:** 1Pharmaceutical Science, Chicago College of Pharmacy Downers Grove, Midwestern University, Downers Grove, IL 60515, USA; 2Pharmazz Inc. Research and Development, Willowbrook, IL 60527, USA

**Keywords:** endothelin B receptors, neurological diseases, neurovascular disorders, sovateltide, IRL-1620, neurogenesis, stem/progenitor cells, regeneration

## Abstract

Neurological/neurovascular disorders constitute the leading cause of disability and the second leading cause of death globally. Major neurological/neurovascular disorders or diseases include cerebral stroke, Alzheimer’s disease, spinal cord injury, neonatal hypoxic-ischemic encephalopathy, and others. Their pathophysiology is considered highly complex and is the main obstacle in developing any drugs for these diseases. In this review, we have described the endothelin system, its involvement in neurovascular disorders, the importance of endothelin B receptors (ETBRs) as a novel potential drug target, and its agonism by IRL-1620 (INN—sovateltide), which we are developing as a drug candidate for treating the above-mentioned neurological disorders/diseases. In addition, we have highlighted the results of our preclinical and clinical studies related to these diseases. The phase I safety and tolerability study of sovateltide has shown it as a safe and tolerable compound at therapeutic dosages. Furthermore, preclinical and clinical phase II studies have demonstrated the efficacy of sovateltide in treating acute ischemic stroke. It is under development as a first-in-class drug. In addition, efficacy studies in Alzheimer’s disease (AD), acute spinal cord injury, and neonatal hypoxic-ischemic encephalopathy (HIE) are ongoing. Successful completion of these studies will validate that ETBRs signaling can be an important target in developing drugs to treat neurological/neurovascular diseases.

## 1. Introduction

Sovateltide or IRL-1620 is a synthetic analog of endothelin 1 (ET-1). ET-1 is an agonist of both endothelin A (ETARs) and endothelin B (ETBRs) receptors, whereas sovateltide is a selective agonist of ETBRs. The endothelin family consists of endothelins (ETs) and their receptors. ETs are potent vasoconstrictor peptides and play important roles in regulating systemic and peripheral vascular systems. The first ET peptide was discovered and isolated from the culture supernatant of porcine endothelial cells by Yanagisawa et al. in 1988 [1]. Its striking vasoconstriction activity emphasized its importance and fueled the ET research in cardio- and neurovascular fields. Soon after the discovery of the first ET, the existence of two other genes encoding ET-like peptides was reported. Based on the order of their discovery, they were named ET-1, ET-2, and ET-3. Each of the three isoforms of ETs consists of 21 amino acids, expressed in various tissues, and plays significant physiological roles in normal and disease conditions. ETs are synthesized as inactive precursor peptides, prepro-ETs, which are proteolytically processed into intermediate 39 amino acid long peptides, Big-ETs (or Pro-ETs). These peptides are finally cleaved to produce ET peptides of 21 amino acids by endothelin-converting enzymes (ECEs) or chymase [2].

ETs are known to bind two different types of receptors, ETARs and ETBRs, with varied affinities. The binding affinity of ETARs to ET subtypes is ET-1 ≥ ET-2 >> ET-3, whereas ETBRs have an equal affinity to each of the subtypes, ET-1 = ET-2 = ET-3 [3,4]. ET isoforms are important and play various pathophysiological roles; however, ET-1 is the most well-studied and known to induce both ETARs and ETBRs mediated signaling. They are expressed ubiquitously in the vascular system and play various physiological roles in a highly distinctive manner in different organs. They are also known to enhance the conversion of angiotensin I to angiotensin II in endothelial cells by stimulating the release of renin and aldosterone [5] and regulating peripheral vascular resistance and blood pressure. The ET system’s complex mechanism of action helps it to perform in an organ-specific manner. The complexity of the ET system is attained through various mechanisms such as different levels of ET production, varied expression ratio ETARs/ETBRs, and their hetero- and homo-dimerization in different organs [6].

Both ETARs and ETBRs, belong to the 7-transmembrane rhodopsin-like (or type A) G-protein coupled receptor family (GPCRA). The GPCRA family includes a wide variety of proteins, including hormones, photoreceptors, neurotransmitters, etc., with different amino acid sequences and functions and transduce extracellular signals through interaction with guanine-binding (G) proteins [7,8,9]. The other GPCR families include secretin-like GPCRs (GPCRB), metabotropic glutamate receptor family (GPCRC), fungal mating pheromone receptors (GPCRD), cAMP receptors (GPCRE), and frizzled/smoothened (GPCRF) [10,11,12,13]. Altogether the GPCRs are on top as a drug target, and hence they are one the most considerable research subjects for the discovery of novel drugs. Approximately 800 GPCRs for endogenous ligands have been reported in humans and mice [14]. Identification of these receptors is based on the DNA sequence, not on their ligands, and therefore more than 140 GPCRs, which are without known ligands, are designated as orphan GPCRs or unclassified GPCRs [15].

ETARs and ETBRs, both receptors, are widely distributed in the whole body, with ETARs mainly located on smooth muscle cells while ETBRs on endothelial cells lining the vessel wall. ETARs are known to play important roles in blood pressure regulation by vasoconstriction and retention of sodium and cell growth by inducing various mitotic pathways, including mitogen-activated protein (MAP) kinases and epidermal growth factor receptors [16,17]. On the other hand, ETBRs are known to be involved in vasorelaxation through vascular endothelial cells and vasoconstriction through vascular smooth muscle cells, regulation of cell growth, and the clearance of ET-1, particularly in lungs where 80% of circulating ET-1 gets cleared [18]. Apart from the vascular system, they are also expressed in various other cells, including epithelial cells and cells of the central nervous system (CNS)—glial cells and neurons [19]. The existence of the ET system in a widespread manner in the CNS is associated with their broader roles in regulating the sympathetic nervous system, blood pressure, blood flow, apoptosis, cell survival, proliferation, and migration in the CNS [20,21,22,23]. Its roles have been demonstrated in the development as well as the pathophysiology of the cerebrovascular system [24,25]. Expression of ETARs and ETBRs varies in different organs; ETARs in the heart and lungs are higher but lower in the CNS, brain, kidneys, lungs, and liver than ETBRs [26,27]. The human brain has been reported with a high density of ETBRs, and they account for about 90% of total ET receptors in the brain [28], while ETARs constitute the smaller portion and are mainly located in the smooth muscle cells of the cerebral tissues [26,29,30]. Several agonists and antagonists of ETARs and ETBRs are continuously being tested for the development of drugs for CNS disorders/diseases. ET isoforms and analogs are an obvious choice for these translational studies, hoping to treat diseases/disorders such as stroke, Alzheimer’s disease, spinal cord injury, etc. [2,31].

Our group and others have studied ETBRs and their roles in neurovascular development and neural diseases. The ontological studies have demonstrated the role of ETBRs on neuronal proliferation and migration and in the expression of angiogenic growth factors in developing mammalian CNS [21,32]. They are critical for the development at prenatal and postnatal stages of life; their deficiency is known to cause CNS congenital anomalies in animal models, whereas their deficiency at the postnatal stage results in increased apoptosis and decreased neuronal progenitor cell (NPC) population in the brain [20,21,33]. They also play crucial roles in neurogenesis and angiogenesis in the CNS in mature animals. Its stimulation following damage to the CNS has also been shown to enhance tissue repair [24,34,35,36,37]. We have mainly used IRL-1620 (INN-sovateltide) as the agonist of ETBR in our research because of its high selectivity. Sovateltide is a synthetic analog of ET-1 and is also known as PMZ-1620 or SPI-1620 or succinyl-(Glu(9), Ala(11,15))-endothelin-1 (8–21). It is a 15 amino acid long chain (Suc-Asp-Glu-Glu-Ala-Val-Tyr-Phe-Ala-His-Leu-Asp-Ile-Ile-Trp) (Figure 1) and has very high selectivity for ETBRs with Ki values 0.016 and 1900 nM at ETBRs and ETARs receptors, respectively. This review briefly describes our preclinical and clinical research of sovateltide mediated ETBRs agonism in diseases such as cerebral stroke, Alzheimer’s disease, spinal cord injury, and neonatal hypoxic-ischemic encephalopathy (HIE).

## 2. Toxicological Studies

Female Swiss Webster mice and Sprague Dawley rats administered with a single (I.V.) dose of sovateltide at the dose levels of 30 µg/kg, 60 µg/kg, 120 µg/kg, and 240 µg/kg showed no mortality. Repeated intravenous administration of sovateltide in mice at the dose levels of 1.0 µg/kg, 2.5 µg/kg, and 6.3 µg/kg for 30 days showed no mortality at any dose levels. The single daily intravenous administration of sovateltide to Wistar rats for 14 days at dose levels of 0, 1, 5, and 15 µg/kg body wt. did not affect survival parameters. No adverse effects were observed at 15 µg/kg/day. Daily intravenous administration of sovateltide in beagle dogs up to 5.0 µg/kg for 14 days showed no adverse effects. No sensitization or allergic potential for sovateltide was observed in guinea pigs. Sovateltide did not affect male fertility of Sprague Dawley rats when administered by an intravenous route at up to 15 µg/kg body wt. Sovateltide, when administered intravenously to Sprague Dawley female rats, exhibited no maternal–fetal toxicity, and it is not teratogenic up to 15.0 μg/kg body wt. Sovateltide, when administered intravenously to New Zealand White female rabbits, exhibited no maternal–fetal toxicity, and it is not teratogenic up to 15.0 µg/kg. Intravenous administration of sovateltide up to 2.0 µg/kg birth weight to conscious, freely moving beagle dogs did not induce any potential effects in any cardiovascular parameters. A bacterial Reverse Mutation Test was performed to evaluate the mutagenic potential of PMZ-1620 using Salmonella typhimurium strains. PMZ-1620 is non-mutagenic as it does not induce (point) gene mutations at histidine locus by base pair changes or frame-shift in the presence and absence of metabolic activation systems in all the five tester strains of Salmonella typhimurium TA1537, TA1535, TA98, TA100, and TA102 up to the concentration of 100 µg/plate. A Mammalian Erythrocyte Micronucleus Test was conducted to assess the clastogenic potential of PMZ-1620 in mice. PMZ-1620 does not cause micronucleus induction in male and female animals treated up to the dose level of 100 µg/kg body wt., as determined by the micronucleus test in the bone marrow cells of the mouse. PMZ-1620 did not cause tumor induction or promote the growth of a tumor. However, when given as adjuvant with chemotherapeutic agents, the complete recession of the tumor was observed.

## 3. Neurovascular Diseases and Clinical Development of Sovateltide

### 3.1. Stroke

Cerebral stroke due to perturbation or blockage of blood flow in brain tissues leads to damage or death of neuronal cells and causes morbidity and mortality in patients. According to recent statistical data, stroke is the second leading cause of death worldwide, with an annual mortality of 5.5 million [38]. Data also suggest that ischemic stroke is more predominant, with ~80% of stroke cases being of the ischemic type [39]. Despite high mortality, treatment of stroke remains challenging. At present, only one FDA-approved drug for ischemic stroke, the thrombolytic agent, rtPA or Alteplase, is available. However, its use is limited due to a short therapeutic window of 3–4.5 h from the onset of symptoms and also has a risk of intracranial hemorrhage in 2–7% of patients [40]. There is an urgent need to develop new effective drugs for stroke to alleviate neurological deficit and repair cerebral damage. However, due to the complex pathophysiology of stroke, the development of new drugs seems quite challenging, as evident by the failure of several drugs in their late stages of clinical trials [41]. Cerebral ischemic pathophysiology involves hypoxia, vascular damage, inflammation, apoptosis, and other events, which cause neuronal cell damage/death and functional impairment in the brain. Significant efforts are being made to understand the complex pathophysiology of stroke, and various mechanisms, e.g., anticoagulation, neuroprotection, and neuroregeneration, are being explored [25]. Interestingly, ET signaling in the mammalian neurovascular system regulates these important pathophysiological events. Therefore, ETs and their receptors, ETARs and ETBRs, are considered one of the most potent targets for developing new drugs for stroke.

Studies have shown an elevated level of ET-1 in blood and brain tissues following cerebral ischemia and correlated with the tissue damage in the ischemic brain [42,43]. Since ET-1 primarily binds to ETARs and acts as a vasoconstrictor, vasoconstriction and associated events were presumed to be the leading cause of cerebral tissue damage, and it was hypothesized that ETARs antagonists would reduce the damage. Therefore, several studies focused on testing the effect of various ETARs inhibitors such as BQ123, SB234551, A-127722, and S-1039 were performed and demonstrated a reduction in neural tissue damage and inflammation and neurological deficits following cerebral ischemia [44,45,46,47,48]. However, unfortunately, none of these could advance to the clinical testing level. Similarly, studies performed using combined ETARs/ETBRs receptors antagonists, TAK-044, bosentan and SB209670 showed mixed results as follows: TAK-044 was effective in decreasing oxidative stress and ischemia, whereas bosentan and SB209670 had no significant effect on improving the condition in animal models of acute ischemic stroke [49,50].

Interestingly, on the other hand, antagonizing ETBRs had deleterious effects and caused poor outcomes [51,52]. These observations suggested the critical roles of ETBRs in protecting brain tissues from ischemic damage following a cerebral stroke. Hence, testing the effect of enhanced ETBRs signaling in ischemic cerebral tissues using specific agonists would be useful for discovering novel potent drugs to treat ischemic stroke.

Studies have shown that ETBRs play an important role in neural cell survival and proliferation [20,53,54]. Normally, expression of ETBRs in the adult is observed on vascular endothelial and smooth muscle cells; however, under pathological conditions, it is also observed in NPCs, NPs, GPs, and glial cells of the central nervous system (CNS) [55,56,57]. Moreover, they also play an essential role in the development of CNS. The rodent model deficient in ETBRs during the prenatal period had CNS disturbances and fatal congenital anomalies. The deficiency of ETBRs in the postnatal stage is known to cause increased apoptosis and decreased neuronal progenitor cell population in different regions of the adult brain [20,21,58]. Besides their importance in brain development, we have demonstrated the role of ETBRs in the repair and regeneration of adult brains after stroke [25,32,59]. We stimulated ETBRs by intravenous administration of a selective agonist sovateltide in permanent middle cerebral artery occluded (MCAO) rats. We observed significantly improved neurological and motor functions and several indications of repair and regeneration, including a decrease in infarct volume and oxidative stress, increased pro-angiogenic, pro-survival, and anti-apoptotic markers, and an increased number of proliferating cells in the brain [24,36,37,60,61,62]. However, these improvements were abrogated by BQ788, an antagonist of ETBRs, which proved the role of sovateltide mediated selective stimulation of ETBRs in these improvements. These results showed the role of sovateltide in providing protection and functional improvement of neural cells in the stroked brain. Some of the remarkable preclinical studies in MCAO rats have demonstrated the roles of sovateltide in enhancing cerebral blood flow and decreasing apoptosis of neural cells after ischemic stroke in the brain [36,37,61,62]. In our recent study, we observed increased phospho-Akt level and decreased Bad expression at 7 h post-MCAO, higher level of anti-apoptotic Bcl-2, and lower level of pro-apoptotic Bax in sovateltide treated MCAO rats than control MCAO rats. The study also showed significantly decreased mitochondrial membrane-bound Bax intensity in sovateltide compared to control MCAO rats at days 1 and 7 post-MCAO [61], which indicated inhibition of mitochondrial apoptotic pathway by sovateltide. The cell damage was assessed by TUNEL assay in these rats, which showed lower cell damage in sovateltide treated rats on day 1 and day 7. Moreover, significantly improved neurological and motor function in these rats was observed after sovateltide treatment [61]. Since the improvement in these functions is known to be associated with the fate of mitochondria and their activity, in another study, we examined mitochondrial fusion, fission, morphology, and biogenesis in sovateltide treated MCAO rat brains. We observed improved mitochondrial fusion, decreased fission, increased size, and biogenesis in sovateltide treated rat brain tissues [63]. We also studied the role of sovateltide mediated ETBRs agonism in the regeneration of neuronal cells in adult MCAO rat brains [63,64] and showed a novel role of sovateltide in neuronal progenitor cells (NPCs) mediated regeneration and repair in MCAO rat brains. Ischemic stroke was induced in the right cerebral hemisphere of adult rats using the MCAO technique, and rats were treated with sovateltide (5 µg/kg body wt.) at 4, 6, and 8 h on day 0, 3, and 6 post-MCAO. They were sacrificed at 24 h or at day 7 post-MCAO, and tissues from the right (RH) and left (LH) cerebral hemispheres were analyzed for expression of neural progenitor markers (DCX, HuC/HuD, and NeuroD1) and stem cell markers (Oct-4 and Sox-2). Significant upregulation of DCX and HuC/HuD in RH, while NeuroD1 in both RH and LH was observed 24 h post-MCAO in sovateltide treated rats. On the other hand, insignificant change in the expression of multipotent stem cell markers, Oct4 and Sox-2 was observed in these rats. Sovateltide treated rats also significantly decreased in the infarct area at day 7 post-MCAO [64].

To explore the role of sovateltide on NPCs differentiation, we cultured adult rat NPCs and exposed them to in vitro hypoxia. NPCs exposed to sovateltide showed significantly higher expression of neuroD1 and NeuN than cells exposed to the vehicle [63]. These results indicated that sovateltide promoted differentiation/maturation of neuronal precursors (NPs) to generate a mature neuronal cell population, which would heal the stroked brain more efficiently (Figure 2). These findings have demonstrated a novel mechanism of action of sovateltide, which promotes differentiation of neuronal progenitors in the ischemic stroked brain and helps in neural regeneration and repair. No other agents in the pipeline of drug development for stroke have shown such type of mechanism of action, and therefore sovateltide has the potential to be developed as a novel first-in-class drug for ischemic stroke.

Clinical development—Our preclinical studies have demonstrated that sovateltide mediated ETBR stimulation in ischemic stroke brain is highly beneficial, and its testing in clinical trials would be useful for the development of sovateltide as a novel agent for the treatment of stroke. We are developing sovateltide for human use; therefore, we carried out extensive toxicological studies in rodents and dogs. After confirming safety in toxicological studies, we performed an open-label phase I study to determine the safety, tolerability, and pharmacodynamics of multiple ascending doses of sovateltide in healthy male volunteers (CTRI/2016/11/007509) (Table 1). The phase I trial was focused on dose escalation design, defining dose-limiting toxicity, and generating important pharmacokinetic and biomarker data critical for determining the dosing strategy in future phase II trials. A human equivalent dose of 0.80 μg/kg body wt. of sovateltide was calculated based on the dose of 5 μg/kg body wt. of sovateltide used in efficacy studies in rats. The phase I study was designed with three cohorts, each cohort had three subjects, and each subject received three doses of either 0.3, 0.6, or 0.9 μg/kg administered at an interval of 4 h as an intravenous bolus over 1 min. None of the subjects experienced serious adverse effects in any cohort; however, nausea and vomiting occurred with the highest dose of 0.9 μg/kg, which resolved within minutes without any intervention. Sovateltide had no significant effect on vital signs, ECGs, or laboratory parameters of healthy male volunteers, and it was well-tolerated and found safe. Based on these results, the Minimum Intolerable Dose (MID) and Maximum Tolerated Dose (MTD) were established as 0.9 μg/kg and 0.6 μg/kg of body wt., respectively [25]. After completion of phase I, we performed a phase II trial to investigate the safety, tolerability, and efficacy of sovateltide in human patients with cerebral ischemia. The proposed therapeutic dose of sovateltide in the phase II trial was 0.3 μg/kg body wt. (lower than the established MTD). The phase II study (CTRI/2017/11/010654; NCT04046484) was conducted in 40 cerebral ischemic stroke patients. A total of 36 patients completed a 90-day follow-up. Eligible human subjects were identified for the trial based on inclusion and exclusion criteria. Some of the inclusion criteria were as follows—adult males or females aged 18 years to 70 years, new (first time) cerebral ischemic strokes subjects presenting up to 24 h, ischemic in origin, no hemorrhage, receiving thrombolytic therapy and standard of care, while exclusion criteria included: subjects receiving endovascular therapy; presenting with lacunar; hemorrhagic and/or brain stem stroke; episode of congestive heart failure; cardiac surgery involving thoracotomy; acute myocardial infarction and arrhythmia; pregnancy, breastfeeding or positive pregnancy test; concurrent participation in any other therapeutic clinical trial, etc. All patients were given standard treatment and care and randomly assigned to either control (saline; *n* = 18; 11 males and 7 females) or sovateltide cohort (*n* = 18; 15 males and 3 females). Clinical outcome parameters National Institute of Health Stroke Scale (NIHSS), Modified Rankin Scale (mRS), and Barthel Index (BI) for cerebral stroke were determined. All the patients received saline or sovateltide between 8 and 24 h after the onset of stroke; however, the number of patients receiving investigational drug within 20 h of onset of stroke was 14/18 in the saline group and 10/18 in the sovateltide group. The baseline characteristics and standard of care (SOC) in both cohorts were similar [65].

This was a prospective, multicenter, randomized, placebo-controlled, double-blinded, exploratory phase II clinical study, and sovateltide was administered as an addition to SOC in patients with acute ischemic stroke. The primary objective of the study was to evaluate the safety and tolerability of sovateltide, while the secondary objective was its efficacy testing on neurological improvements using the NIHSS, mRS, and BI scales and quality-of-life assessments using the EQ-5D [66,67] and the SSQoL [68]. Sovateltide was found to be well-tolerated, and no adverse events were seen in patients who received all nine doses of sovateltide. Sovateltide treatment led to improvements in mRS and BI scales on day 6 compared to day 1, indicating quicker recovery of patients. It also increased the frequency of favorable outcomes at day 90; 56% of patients in the sovateltide group had an improvement of ≥6 points in NIHSS, while 43% of patients in the saline group had such improvement. Albeit statistically, the data was not significant, the trend of better improvement in “favorable outcomes” in sovateltide patients compared to saline was seen. An improvement of ≥2 points in the mRS was observed in 60% of patients in sovateltide and 40% of patients in saline groups. It was statistically significant at 90% confidence level; however, significance was lost at 95% confidence level. A significant BI improvement of ≥40 points was reported in 64% of sovateltide patients and 36% of saline patients (*p* = 0.0112). The number of patients with complete recovery achieving an NIHSS score of 0 and a BI score of 100 was significantly higher (*p* < 0.05) in the sovateltide cohort compared to that of saline at the 95% confidence level. More patients in the sovateltide group had an mRS score of 0 (*p* = 0.1193), but this was not statistically significant. These results showed a better trend toward the complete recovery of patients after sovateltide treatment than saline, although some of the clinical outcome scales did not reach statistical significance. No drug-related adverse events were reported, and the results indicated a clear superiority of sovateltide over SOC, resulting in better clinical outcomes for patients with acute cerebral ischemic stroke (Table 1).

Overall, the phase II study has demonstrated sovateltide as a safe and well-tolerated agent in patients with acute ischemic stroke. In addition, the intravenous administration of sovateltide provides significant improvement in several but not all parameters of neurological outcomes in stroke patients. Our phase II study was an exploratory type because using sovateltide is an entirely new approach in the treatment of stroke, and thus far, it is the only study to assess its effects on neurological outcomes in cerebral ischemic stroke patients.

The promising results of both phase I and phase II studies encouraged us to further investigate the efficacy of sovateltide in human phase III study in patients with cerebral ischemia. We started a randomized and parallel-assigned phase III trial (NCT04047563) in November 2019. It is expected to enroll 158 patients randomly assigned to receive three doses of sovateltide via intravenous bolus over a 6-day period plus best available SOC, or three doses of equal volume normal saline plus best available SOC over the same period. Sovateltide will be administered as an intravenous bolus over 1 min at every 3 ± 1 h on days 1, 3, and 6 (total dose/day: 0.9 µg/kg body wt.). Adults aged 18 to 76 with ischemic stroke and who have prior radiology of the stroke confirmed by a computed tomography (CT) scan or magnetic resonance imaging (MRI) are included. Additionally, patients must receive the first dose of sovateltide within less than 24 h of stroke onset. Exclusion criteria include patients receiving endovascular therapy, patients who are classified as comatose, or those who show evidence of intracranial hemorrhage. The primary outcome measures include a change in the National Institute of Health Stroke Scale (NIHSS) and a change in modified Rankin Scale (mRS) score, among others. The phase III trial completed the enrollment of 158 adult acute ischemic stroke patients in February 2022 [69].

### 3.2. Alzheimer’s Disease

Alzheimer’s disease (AD) is a neurodegenerative disease characterized by progressive loss of neuronal cells and cognitive decline and is also associated with aging [70]. According to a recent report, approximately 5.8 million Americans of all ages were suffering from AD in 2019, which is projected to rise to 14 million by 2050 [71]. The cost in 2019 for all individuals with AD and other dementias was approximately USD 277 billion. Compared to other diseases such as stroke and heart failure, the proportion of deaths related to AD is going up, and it increased by 89% between 2000 and 2014 [72,73]. At present, the management and care of AD rely only on two classes of pharmacologically approved therapies: first, cholinesterase inhibitors, and second, N-methyl-D-aspartate receptor antagonists. The cholinesterase inhibitors donepezil, rivastigmine, and galantamine are recommended therapy for patients with mild, moderate, or severe AD dementia [73]. On the other hand, memantine is approved for use in moderate-to-severe AD patients. It has antagonistic activity on N-methyl-D-aspartate receptor and agonistic activity on dopamine. Although the existing drugs help in concealing AD symptoms, they fail to cure or delay AD progression. Some of the pathological hallmarks of AD include β-amyloid plaque deposition and neurofibrillary tangles of hyperphosphorylated tau protein [74,75]. The neuropathology involved in AD is highly complex, making drug development very difficult, as evident by failing most of the AD drug trials in the past. Approximately, 413 AD trials were performed from 2002 to 2012 but 99.6% failed and could not meet the endpoints [76]. One of the reasons for the failure of these trials could be their reliability on the most common hallmarks of the AD pathology, beta-amyloid (Aβ) and tau, which are still debatable whether they have any causative role in AD pathology. At present, immunotherapy targeting the clearance of Aβ fibrils and plaques is being developed, which will help in understanding the role of AD hallmarks in the disease. Some of the drugs in this category are aducanumab, donanemab, bapineuzumab, solanezumab, gantenerumab, crenezumab, and ponezumab (PMID: 24959143, PMID: 33720637). Unfortunately, the clinical trials of aducanumab (Aduhelm) failed to demonstrate its efficacy in slowing memory loss or cognitive decline. However, its controversial approval using the FDA’s Accelerated Approval Pathway, based on Aβ plaque clearance, which was a surrogate endpoint of the trial has deeply engraved the AD drug development [77,78,79]. Hence, for developing effective AD drugs, more research in the future focused on examining the neural pathophysiological pathways essential for neuronal cell survival, regeneration, and functional improvement in adult brains are required. Several studies carried out by our group and others have shown that the ET system is involved in pathways related to neural cell survival, regeneration, and repair of adult brains after damage [36,37,60,61,63,64,65]. Therefore, testing its potential as a drug target in AD would be useful in developing a new drug, which could repair the damaged area in the brain and restore the cognitive ability and memory of AD patients.

Several studies have demonstrated the involvement of the ET system in AD and provided the following evidence. Significantly elevated levels of ECE-2 and ET-1 have been correlated with AD disease progression and in vitro and in vivo studies have demonstrated their abnormal production because of Aβ [80]. Gene mutation analysis study has shown increased production of Aβ_1–40_ and Aβ_1–42_ after deletion mutations in either ECE-1 or ECE-2 in mouse brains [81,82]. Mice with ECE-2 knockout were observed with impaired learning and memory [83]. Thus, studies have demonstrated a direct link between ETs and AD pathogenesis, and a relationship between ECEs and Aβ protein turnover in the adult brain [80,84]. Apart from the above-mentioned direct links, AD development and progression are also associated with regionally reduced cerebral blood flow and vascular dysfunction [85], which are regulated by ETs [54,86,87]. The ET-1 mediated vasoconstriction in middle cerebral and basilar arteries has been found to be increased following exposure to Aβ [88], also an elevated level of ET-1 has been detected in the post-mortem brains of individuals with AD compared to non-AD individuals. These observations indicated the role of ET-1 in neuronal dysfunction in the early stage of AD development and progression.

We initially hypothesized a potential benefit of ETARs antagonists to reduce the ET-1 mediated effects in the AD brain, which could be developed as novel agents for AD treatment. Our group examined the effect of ETARs antagonists on Aβ-induced neuronal damage and cognitive dysfunction in an adult rat model of AD. We used BQ123 and BMS182874, specific ETARs antagonists, in our studies. Both BQ123 and BMS182874 significantly improved the spatial memory deficit and reduced oxidative damage in the brain caused by Aβ. A significantly reduced escape latency and increased preference for the target quadrant were observed in BQ123, or BMS182874 treated AD rats. However, rats treated with a nonspecific ETARs/ETBRs receptors antagonist, TAK-044, had no improvement in spatial memory deficit [89]. Lack of improvement with nonspecific ETARs/ETBRs antagonist indicated a role of ETBRs signaling in the AD brain. Since ET-1 could bind to both ETARs and ETBRs receptors, normally ETBRs might be countering the action of ETARs signaling; however, in the presence of ETARs antagonists, they would be potentiating their signals. These results suggested the important roles of ETBRs receptors in AD development and progression and could be an appropriate target for AD drug development. ETBRs have shown anti-apoptotic activity in the protected cells from neurotoxicity of Aβ in cultured neurons [21,24,25,90]. Cortical neural progenitor cells showed a high level of ETBRs, and their stimulation leads to proliferation and migration, indicating that stimulation of ETBRs enhances neuroregeneration by directly acting on neural progenitors [20,91,92]. Stimulation of ETBRs is also known to elicit vasodilatation, and previous studies in our laboratory have demonstrated that intravenous administration of sovateltide (an ETBRs agonist) increased cerebral blood flow in normal rats [93] and that the expression of an anti-apoptotic marker, Bcl-2, was found to be increased, and pro-apoptotic marker, Bax was found to be decreased in the neuronal cell line, PC-12 [94]. While, in a rat model of AD produced by injecting Aβ_1–40_ intracerebroventricularly, sovateltide (5 μg/kg body wt.) treatment resulted in a significant functional recovery. We examined their behavioral and spatial memory impairments using a Morris Water Maze. AD rats showed impairment in spatial memory as evidenced by significantly longer escape latencies and no preference for the quadrant, which previously contained the platform in the probe trial [95], while AD rats treated with sovateltide significantly reduced the spatial memory deficit caused by Aβ. The Aβ plaque is known to induce oxidative damage to neural cells, and we examined whether sovateltide treatment had any effect on the oxidative stress in the brain of AD rats. Sovateltide treated rats showed significantly decreased levels of malondialdehyde (MDA) and increased levels of reduced glutathione (GSH) and superoxide dismutase (SOD), which indicated significantly reduced oxidative stress in AD rats after sovateltide treatment. However, these improvements were blocked when animals were administered with ETBRs antagonist BQ788, which confirms the effects were due to selective stimulation of ETBRs by sovateltide in these AD rats [95].

Our other studies related to cerebral stroke in the MCAO rat model of stroke have shown that sovateltide treatment could increase NGF and VEFG expression in rat brains after ischemia [34]. Since these factors are important for neurogenesis, an increase in the expression of these markers suggested the role of ETBRs in neurogenesis in damaged brain tissues. Therefore, we treated an APP/PS1 mouse AD model with sovateltide and assessed neurogenesis in the AD mouse brain. The treated mice were observed with a significantly increased expression of neural progenitor markers, NeuroD1 and Doublecortin (DCX), along with elevated expression of nuclear NeuN (a marker for mature neurons) in the mouse brain. These observations suggested an increase in neuronal progenitors and their differentiation to produce mature neuronal cells expressing NeuN in sovateltide treated mice, indicating a higher potential for neuronal regeneration. The synapse formation by newly produced cells in the brain is a prerequisite for its function restoration; hence we assessed the expression of pre-synaptic (synapsin1 and synaptophysin) and post-synaptic marker (PSD95) in these mice. Sovateltide significantly increased the expression of these markers in AD mouse brain tissues. The assessment of learning and memory in control AD mice showed significant impairment in spatial memory as evident by significantly longer escape latencies and no preference for the quadrant which previously contained the platform in the probe trial. On the other hand, AD mice treated with sovateltide had significantly reduced (45, 40, and 46%) learning and memory deficit in 6, 9, and 12 months of age.

Overall, our studies have demonstrated that ETBRs agonist, sovateltide is an effective agent, which can significantly reduce the AD-associated neurodegeneration and help in neuronal regeneration and cerebral function recovery by promoting neuro- and synapto-genesis.

Clinical development—Our preclinical studies have demonstrated that sovateltide mediated ETBR stimulation in AD animal models is highly beneficial, and its testing in clinical trials would be useful for developing a novel agent for the treatment of AD. We produced sovateltide for human use and carried out toxicological studies of sovateltide in mice, rats, and dogs. After toxicological testing, we performed an open-label phase I study to determine the safety, tolerability, and pharmacodynamics of multiple ascending doses of sovateltide in healthy male volunteers (CTRI/2016/11/007509). The phase I trial was focused on dose escalation design, defining dose-limiting toxicity, and generating important pharmacokinetic and biomarker data critical for determining the dosing strategy in future phase II trials. Currently, we are investigating the potential of sovateltide as a novel drug to treat human AD patients in a clinical phase II trial (NCT04052737). Our phase II trial is a prospective, multicentric, randomized, double-blind, placebo-controlled study to compare the safety and efficacy of sovateltide therapy along with standard supportive care in subjects of mild to moderate Alzheimer’s disease. In the treatment group, three doses of sovateltide, at 0.3 μg/kg body wt., are administered as an intravenous bolus over 1 min every 3 ± 1 h (total dose/day: 0.9 µg/kg body wt.). The same dosing regimen is repeated every month for 6 months post-randomization. While in the control group, three doses of an equal volume of normal saline are administered as an IV bolus over 1 min every 3 ± 1 h on day 1 post-randomization. The same dosing regimen is repeated every month for 6 months post-randomization. In both treatment groups, subjects are provided the best available standard of care for AD. The total number of participants in the study is 80. An interim analysis of 62 patients (control *n* = 31 and sovateltide *n* = 31) has shown the effect of sovateltide in slowing down the disease progression in AD patients after 180 days of treatment and follow-up. The change from baseline in the Alzheimer’s Disease Assessment Scale cognitive subscale (ADAS-Cog) was 1.382 ± 0.920 (95% CI −0.702 to 3.466) in control and 0.247 ± 0.955 (95% CI −1.918 to 2.412) in the sovateltide treated patients. The baseline ADAS-cog score was 22.59 and 27.85 in the control and the sovateltide groups, respectively, indicating that disease severity was more in the sovateltide group than in the control group at the start of the treatment. The average baseline score of the Mini-mental state examination (MMSE) was 19.45 and 17.42 in the control group and sovateltide group, respectively. MMSE score changed by 1.226 ± 0.498 (95% CI 0.095 to 2.356) in the control group and 0.9355 ± 0.5004 (95% CI −0.198 to 2.069) in the sovateltide group from their respective baseline value. We observed that Neuropsychiatric Inventory (NPI) score was changed by −0.548 ± 1.123 (95% CI −3.093 to 1.996) from a baseline of 9.516 in the control group, whereas by −1.968 ± 1.077 (95% CI −4.407 to 0.471) from a baseline of 9.871 in the sovateltide group (Table 1). The interim analysis of the phase II trial data indicates a potential beneficial effect of sovateltide in AD patients with late early to moderate stages of the disease [69].

### 3.3. Spinal Cord Injury

The spinal cord with the brain constitutes the central nervous system. It is a long bundle of nerves and cells which extends from the brain to the lower back. It is required to carry signals to and from the brain, including nerve impulses for movement, sensation, pressure, temperature, pain, and many more. Moreover, it is also known to act independently of the brain in conducting motor reflexes. Thus, the spinal cord plays a vital role in the body’s functioning. The damage to the spinal cord is known as spinal cord injury (SCI), which causes temporary or permanent changes in its function. The SCI etiologies could be traumatic or non-traumatic; however, more than 90% of SCI-reported cases are traumatic, caused by tragic incidences such as traffic accidents, violence, sports, or falls [96]. In traumatic SCI, the primary injury damages tissues in the cord and leads to a highly complex secondary injury cascade, including ischemia, inflammation, and the death of neurons and glial cells. Formation of glial scar and cystic cavities are followed, and consequently, changes in the organizational and structural architecture of the spinal cord are resulted, which may cause permanent neurological deficits. The secondary injury also includes hyperinflammatory and cytotoxic conditions, which further damage neuronal cells and inhibit regeneration and repair processes in the damaged spinal cord.

Moreover, the regenerative capacity in the spinal cord is limited because of an abysmal number of neural progenitor cells and restricted plasticity, which makes the intrinsic recovery potential of the spinal cord very poor and rare [97]. Nonetheless, existing therapeutic approaches include early surgery, strict blood pressure control, and treatment with steroids, which are still debatable for their effect, and they are primarily focused on mitigating secondary injury of SCI, not to cure. Hence, currently available SCI interventions have failed to improve the SCI treatment outcomes. Therefore, finding a cure for SCI is urgently required otherwise, it’s devastating physical, social, and vocational consequences for patients and their families would be continued. In the past four decades, numerous therapies aimed to improve neuroprotection and neurodegeneration have shown little promise from the preclinical to the clinical stage of development. However, further research aimed to understand the complex pathophysiological cascade and neuroregeneration and repair in SCI are required to overcome the limitations in translating the SCI research data obtained from animal models to clinical trials.

The SCI clinical trials carried out so far have explored various approaches, including pharmacologic, cell-based, physiologic, and rehabilitation, to reduce secondary injury and overcome barriers of neuroregeneration and recovery. Undoubtedly, several clinical trials have been completed and are currently being carried out; however, discovering a potent drug to cure SCI still looks beyond reach; a brief detail about these trials is described by Donovan et al. [98]. Nonetheless, a tailored treatment plan combining many of these strategies is being emphasized, which would offer significant benefits for persons with SCI. Since regeneration and repair of tissues after damage in organs is the most important for their functional restoration, we believe that the success of the SCI treatment strategy would depend upon how efficiently we could promote the regenerative process in the damaged spinal cord. Hence, we suggest spinal regenerative research should be at the core of every combinatorial tailored plan of SCI treatment, and therefore more research in this direction should be promoted to discover novel spinal cord regenerative therapeutics.

Our research work focused on regeneration and repair of adult brains after stroke and Alzheimer’s disease by utilizing sovateltide to elicit ETBRs in the brain is highly promising, as described in the above sections of the review. The preclinical and clinical studies of sovateltide in these diseases have created hope for developing sovateltide as a first-in-class therapeutic. It works uniquely by promoting differentiation of neuronal progenitor cells and reducing apoptosis and oxidative stress besides increasing blood flow in the damaged brain tissues and helping in protecting cerebral tissues from damage and promoting regeneration and repair in the damaged tissues. The spinal cord is also a part of the CNS and known to express ETBRs; hence we hypothesized that sovateltide could help treat SCI. Receptor binding studies in mammals, including humans, demonstrated that ETARs and ETBRs are distributed throughout the spinal cord [99], and their signaling play critical roles in the inflammatory response and oxidative stress, which are known to affect the neurological recovery after SCI mediated disruption of blood–spinal cord barriers [100,101].

Furthermore, it has been shown that blocking ETARs could significantly reduce the expression of inflammatory factors, e.g., TNF-α, IL-1β, and IL-6 [102]. On the other hand, using a nonselective inhibitor of ET receptors, bosentan inhibited SCI-induced pain response probably by blocking both ETARs and ETBRs [103]. At the same time, the use of SB209670, an ETARs/ETBRs antagonist, to the lesion site of SCI showed the effect on reducing axonal damage after injury [104]. These reports have provided evidence that ETARs, ETBRs, and endogenous ETs play an important role in SCI pathophysiology. Therefore, they could be potential therapeutic targets for the regenerative drug development for SCI to alleviate the effect of primary and secondary injuries, which may ultimately help cure SCI.

Although these studies have shown the involvement of the ET system in SCI, they were based on pharmacologically stimulating or blocking of ETARs or ETARs/ETBRs, and none examined the effect of selectively stimulating ETBRs after SCI. We and others have shown the role of ETBRs in reducing oxidative stress, increasing blood flow and angiogenesis, enhancing the proliferation of neuronal progenitors and protecting cerebral cortical neurons against apoptosis [20,105,106]. Moreover, our findings have shown neuroregeneration and functional recovery following selective ETBRs stimulation by sovateltide in animal models of cerebral ischemia and Alzheimer’s disease [34,63,64]. Since most of the events and factors are common in cerebral and spinal cord injury, such as oxidative stress, inflammation, and neuronal cell death are known to play a critical role in both cases. Like cerebral neuronal regeneration, the generation and proliferation of spinal cord neurons are also essential for functional recovery of the spinal cord after injury. Therefore, we hypothesized that stimulation of ETBRs using sovateltide in the spinal cord would be useful in alleviating the damaging effects and promoting neuronal regeneration and repair after SCI.

Our study used adult Sprague Dawley rats; they were subjected to a moderate spinal contusion of 150 kdyn at the thoracic level (T10) and treated with 1, 3, or 5 μg/kg sovateltide or with the same volume of saline. Three doses of sovateltide were injected intravenously at 2 h intervals on days 1, 3, and 6 post-injury in test rats. The same volume of saline was injected in control rats. Motor functions were determined by the Basso, Beattie, Bresnahan scale. We observed significantly improved motor functions in the hind limb of rats treated with sovateltide compared to saline-treated animals (unpublished observation). We also observed increased expression of ETBRs and synapsin in the spinal cord of rats following SCI. Our investigation at the cellular level on day 60 post-trauma showed a significantly higher number of NeuN-positive neurons and BMP-positive myelinated fibers in the proximity of lesions in animals treated with sovateltide compared to saline. Moreover, we also found that sovateltide enhanced stem cell self-renewal by increasing the expression of SOX-2, Nanog, OCT-4 and activating the PI3K-AKT pathway in the murine spinal cord explants grown for 1–3 days ex vivo. Overall, these results have indicated the potential of sovateltide in attributing to neuroregeneration, synaptic remodeling, and improving motor functions after SCI. After encouraging results from these studies, we are testing the potential of sovateltide to treat SCI in human clinical trials (phase I—CTRI/2016/11/007509, completed; phase II—NCT04054414, ongoing), which will help us in developing sovateltide as a novel drug for SCI. The phase II study is prospective, multicentric, randomized, double-blind, parallel, saline controlled to compare the safety and efficacy of sovateltide therapy and standard supportive care in patients with acute spinal cord injury. Sovateltide dosing strategy is as follows—administration of three doses of sovateltide (0.3 μg/kg body wt.) as an intravenous bolus over 1 min at every 3 ± 1 h on day 1, 3, and day 6 (total dose/day: 0.9 µg/kg body wt.). In the control group, three doses of an equal volume of normal saline are administered as an I.V. bolus over 1 min every 3 ± 1 h on days 1, 3, and 6 post-randomization. In both groups, subjects are provided the best available standard of care. The trial is being carried out with quadruple (Participant, Care Provider, Investigator, Outcomes Assessor) masking protocols in place and is expected to be completed in the second half of 2022 (Table 1).

Further studies to understand the role of ETBRs in SCI rodent models and anticipated results from the phase II trial will help to elaborate the role of ETBRs as an attractive pharmacological tool for SCI treatment and would establish a new ETBRs based spinal cord regenerative strategy to increase neuronal survival, regeneration, and function after injury.

### 3.4. Neonatal Hypoxic-Ischemic Encephalopathy

Hypoxic-ischemic encephalopathy (HIE) is a major cause of neurologic disabilities and mortality in term neonates, which is mainly caused due to intrapartum complications. Its clinical outcomes are devastating and include neurological disabilities such as cerebral palsy, seizures, and neurodevelopmental disorders in neonates. The incidence of HIE is estimated to be ranged from 1 to 8 per 1000 live births in developed countries and about 26 per 1000 live births in underdeveloped countries [107], constitutes the second cause of mortality in neonates and the third cause of mortality in children < 5 years old [108,109]. The pathophysiology of HIE involves a decrease in placental perfusion or disruption of the delivery of oxygen and glucose in the umbilical cord because of a variety of conditions, including placental abruption, prolapse of the umbilical cord, and uterine rupture. The inadequate placental perfusion causes hypoxia and disrupts the homeostasis in the fetus. Hypoxia leads to a decrease in cardiac output and reduces cerebral blood flow in the fetus. In a moderate decrease, the cerebral blood flow is shunted from the anterior circulation to the posterior circulation, and perfusion in the brainstem, cerebellum, and basal ganglia is maintained. Consequently, the damage is restricted mainly to the cerebral cortex and watershed areas of the cerebral hemispheres. While on the other hand, an acute hypoxia condition invokes an abrupt decrease in cerebral blood flow and produces injury also in the basal ganglia and thalami [110]. The acute hypoxic damage in HIE has been categorized in different phases based on the temporal sequence of the injury. The reduced delivery of oxygen and glucose leads to anaerobic metabolism in the brain, which causes decreased ATP and increased lactic acid production. Due to ATP shortage, transcellular transport is reduced, which causes intracellular accumulation of sodium, water, and calcium. Moreover, upon membrane depolarization, cells release the excitatory amino acid glutamate, and more calcium flows into the cell via N-methyl-D-aspartate–gated channels. The high calcium level in cytosol induces an array of deleterious effects, including necrosis or calpain activation followed by apoptosis, also known as excitotoxicity. Production of free radicals is known to be increased in the hypoxic condition, which causes peroxidation of free fatty acids, which causes more cellular damage. In trauma, neural cells are also known to produce several-fold higher levels of NO, which causes further cellular damage. In summary, the culmination of energy failure, acidosis, glutamate release, lipid peroxidation, and the toxic effect of nitric oxide leads to cell death via necrosis and activates apoptotic cascades in the spinal cord after injury. Depending on the timing of the injury and the available medical intervention, partial recovery may occur during the first 30 to 60 min after the insult. This partial recovery provides a latent phase of injury, which could last from 1 to 6 h and is characterized by recovery of oxidative metabolism, inflammation, and continuation of the activated apoptotic cascades [111]. The secondary phase of injury commonly follows the latent phase within approximately 6 to 15 h after the acute insult. Events in this phase include cytotoxic edema, excitotoxicity, and severe mitochondrial activity failure, leading to cell death and clinical deterioration. The occurrence of seizures is very common in this phase of injury. In some months after the acute insult, a tertiary phase occurs, which involves late cell death, remodeling of the injured brain, and astrogliosis [112]. Neonates with suspected HIE are classified according to the Sarnat staging system [113], which evaluates the level of consciousness, muscle tone, tendon reflexes, complex reflexes, and autonomic function. The Sarnat stage classifies neonatal HIE into the following three categories: stage I (mild), stage II (moderate), and stage III (severe). The pathophysiology of HIE is highly complex, which makes the discovery of effective drugs and interventions very difficult, as evidenced by the failure of several drug trials [114]. At present, there is only one therapy for HIE; therapeutic hypothermia, which has been accepted in its clinical management. The effectiveness of hypothermia in reducing neurological injury caused by HIE has been demonstrated through several clinical trials, including the ICE study [115], Cool Cap study [116], NICHD study [117], and TOBY study [112]. Although the procedure of therapeutic hypothermia is effective, it suffers various limitations, such as the unavailability of trained personnel, equipment, and pediatric neurology support at most of the NICUs (newborn ICUs). Furthermore, its efficacy in preventing neurological disorders in HIE neonates is poor as more than 40% of neonates undergoing hypothermia still develop adverse neurological outcomes [118]. Moreover, the long-term (>2 years) impact of this therapy on neurodevelopment in children remains unclear [119]. Presently, it is a matter of the most concern that a significant number of HIE-affected infants still die or suffer from neurological disabilities whether they receive the hypothermia treatment or not [116,120,121,122]. A decade ago, to address these concerns, the Eunice Kennedy Shriver National Institute of Child Health and Human Development (NICHD) had invited a panel of experts to review the available evidence, identify knowledge gaps, and suggest research priorities. The panel recommended the development of adjuvant therapies to hypothermia, the use of biomarkers, as well as further refinements in therapeutic hypothermia [121]. Hence, HIE has been an area of global health concern, and the development of new drugs/interventions or adjuvants to therapeutic hypothermia is urgently required. Nonetheless, several adjuvant therapies to hypothermia are undergoing and evaluating the improved survival and neurodevelopmental outcomes in newborns with HIE; more research aiming the treatment optimization and prevention and/or eventually reversal of the HIE damage will be useful. One such potential adjuvant or alternative therapy for neonatal HIE could be a selective agonist of ETBRs such as sovateltide. As previously mentioned, our group has demonstrated the role of sovateltide in both neuroprotection and neuroregeneration in adult rat models of cerebral ischemia [37,60]. Additionally, it was found to reduce oxidative stress, increase pro-angiogenic, pro-survival, and anti-apoptotic markers, and increase the number of proliferating cells in rodent brains [36,37,60,61]. Importantly, our recent studies have shown the role of sovateltide mediated ETBRs stimulation on regulating the mitochondria-mediated apoptotic pathway in rat brains after ischemic stroke damage and its correlation with improved neurological and motor functions in these rats [61]. Furthermore, we observed that it improved mitochondrial fusion, decreased fission, increased size, and biogenesis in sovateltide treated rat brains with ischemic stroke [63]. We also demonstrated a novel role of sovateltide in NPCs-mediated regeneration and repair in the rat brain tissues [64]. Therapeutic hypothermia is known to preserve mitochondrial function; it is conceivable that its future adjuvants targeting mitochondria would ensure further efficacy. Hence, it would therefore be of interest to determine the efficacy of sovateltide (which has shown its roles in mitochondrial fate determination as well as in regeneration and repair of neural tissues) in an animal model of neonatal HIE.

## 4. Conclusions

Neurological/neurovascular disorders or diseases have been considered non-curable in the absence of an effective drug till now. Our preclinical and clinical research on these disorders has demonstrated sovateltide as a novel potential therapeutic candidate. We carried out many studies to understand the effects of sovateltide in these disease conditions and explore the mechanism of action in animals’ brains and spinal cord. Sovateltide has been shown to induce ETBRs mediated signaling and significant improvement in various cellular, molecular and physiological events, e.g., anti-oxidative activity, anti-apoptosis, mitochondrial fusion and biogenesis, angiogenesis, tissue re-perfusion, neuronal differentiation, tissue regeneration, and repair in the CNS (the brain and spinal cord) of animal models of stroke, Alzheimer’s disease, neonatal hypoxic-ischemic encephalopathy, and spinal cord injury. Our finding, sovateltide promotes differentiation of neuronal progenitors to produce mature neuronal cells after CNS damage is unique, proving it as a first-in-class drug candidate for treating CNS disorders. Furthermore, sovateltide has been remarkably effective in treating acute ischemic stroke in animal models and human patients in phase II clinical trials and helps in improving neurological and motor functions. Phase III trial of stroke and phase II trials of Alzheimer’s disease and SCI in human patients are currently being conducted (Pharmazz Inc.), while the HIE study is in its preclinical stage. Thus, our studies have demonstrated the highly promising effects of sovateltide on the neurovascular system and improved recovery, comprehensively after various injuries/insults, which have generated immense hope of developing an effective therapy with the potential to treat various neurological disorders.

## Figures and Tables

**Figure 1 ijms-23-03146-f001:**
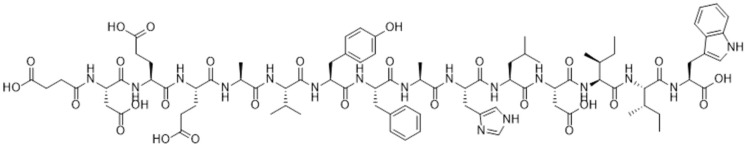
Chemical structure of sovateltide.( Adapted with permission from MedKoo Biosciences, Inc. © MedKoo Biosciences, Inc. https://www.medkoo.com/products/34963, accessed on 10 March 2022).

**Figure 2 ijms-23-03146-f002:**
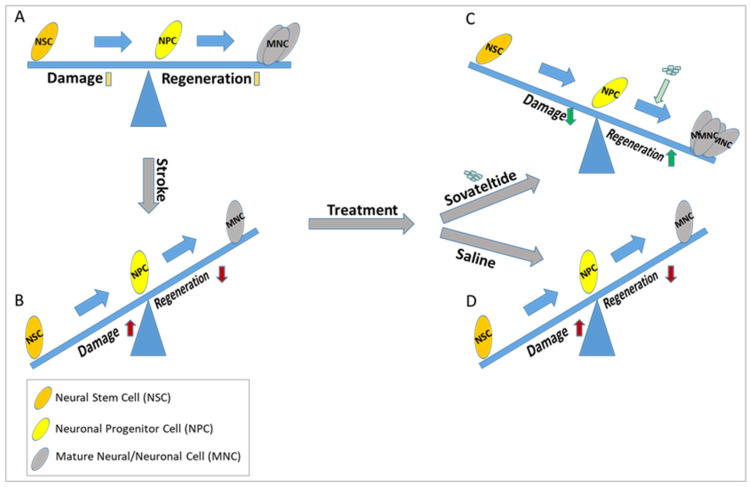
Diagrammatic representation of the effect of sovateltide on neuronal progenitor cell differentiation and neural regeneration after cerebral ischemic stroke. (**A**) Normal homeostasis, (**B**) Higher damage than regeneration (probably due to less differentiation of NPCs), (**C**) Higher regeneration and lower damage after sovateltide treatment (probably due to higher NPCs differentiation), which remains similar after saline treatment (**D**).

**Table 1 ijms-23-03146-t001:** Clinical trials of sovateltide for ischemic stroke, Alzheimer’s disease, and spinal cord injury.

Clinical Trials Involving Sovateltide	Disease	Design	Findings
1. Phase I—CTRI/2016/11/007509	None	Total cohorts—3. Healthy subjects in each cohort—3. Dose—3 doses of either 0.3, 0.6, or 0.9 μg/kg at an interval of 4 h as an intravenous bolus over 1 min in each subject.	Minimum Intolerable Dose (MID) of 0.9 μg/kg and Maximum Tolerated Dose (MTD) of 0.6 μg/kg of body wt. were established. No serious adverse effects in any cohort. No significant effect on vital signs. Sovateltide was found well-tolerated and safe at these doses.
2. Phase II—CTRI/2017/11/010654 and NCT04046484	Ischemic Stroke	Prospective, multicenter, randomized, placebo-controlled, double-blinded, and exploratory clinical study. Total cerebral ischemic stroke patients—40 (36 patients completed 90-day follow-up). Dose—0.3 μg/kg body wt. per dose, 3 doses per day at an interval of 3 ± 1 h as an intravenous bolus over 1 min in each subject (total dose/day: 0.9 ug/kg body wt.). Treatments—Treatments on days 1, 3, and 6 with sovateltide or an equal volume of saline. Sovateltide (*n* = 18; 15 males and 3 females) or saline (*n* = 18; 11 males and 7 females) + standard treatment and care. Clinical outcome parameters—National Institute of Health Stroke Scale (NIHSS), Modified Rankin Scale (mRS), and Barthel Index (BI). Primary objective—Testing safety and tolerability of sovateltide. Secondary objective—efficacy testing on neurological improvements using the NIHSS, mRS, and BI scales, and quality-of-life assessments using EQ-5D and SSQoL.	Improvements in mRS and BI scales in the sovateltide group on day 6 compared to day 1 indicated a quicker recovery in the sovateltide cohort. At day 90, improvement of ≥ 6 points in NIHSS in 56% of sovateltide and 43% of the saline group. A significant improvement (*p* = 0.0112) of ≥40 points in BI in 64% of sovateltide and 36% of saline patients. An improvement of ≥2 points in the mRS in 60% of sovateltide and 40% saline group. Higher number of patients had complete recovery with NIHSS score of 0 (*p* < 0.05), a BI score of 100 (*p* < 0.05) and an mRS score of 0 (*p* = 0.1193) in sovateltide cohort compared to saline. No adverse effects due to treatment were observed. These results showed the safety and efficacy of sovateltide with a better trend toward the complete recovery of ischemic stroke patients compared to saline.
3. Phase III—NCT04047563	Ischemic Stroke	Prospective, multicenter, randomized, placebo-controlled, double-blinded, and exploratory clinical study. Total cerebral ischemic stroke patients—158. Age range—18–76 years. Dose—0.3 μg/kg body wt. per dose, 3 doses per day at an interval of 3 ± 1 h as an intravenous bolus over 1 min in each subject (total dose/day: 0.9 ug/kg body wt.). Treatments—Treatments on days 1, 3, and 6 with sovateltide or an equal volume of saline + standard treatment and care. Clinical outcome parameters—National Institute of Health Stroke Scale (NIHSS), Modified Rankin Scale (mRS), and Barthel Index (BI). Primary objective—Efficacy testing on neurological improvements using the NIHSS, mRS, and BI scales. Secondary objective—Quality-of-life assessments using EQ-5D and SSQoL. Recurrence of ischemic stroke, incidence of intracerebral hemorrhage and mortality, alteration in cognition, and sovateltide related adverse events.	Enrollment of patients is complete. Results are awaited.
4. Phase II—NCT04052737	Alzheimer’s Disease (AD)	Prospective, multicentric, randomized, double-blinded, placebo-controlled study in mild to moderate Alzheimer’s disease to assess safety and efficacy of sovateltide therapy. Total participants—80. Dose—0.3 μg/kg body wt. per dose. Three doses of sovateltide, at every 3 ± 1 h (total dose/day: 0.9 µg/kg body wt.). Repeat doses every month for 6 months. In both treatment groups, subjects are provided the best available standard of care for AD.	An interim analysis of 62 patients (control *n* = 31 and sovateltide *n* = 31) after 180 days of treatment and follow-up. AD Assessment Scale cognitive subscale (ADAS-Cog) changed 1.382 ± 0.920 (from a baseline value 22.59) in control and 0.247 ± 0.955 (from a baseline value 27.85) in the sovateltide treated patients from their respective baseline. Mini-mental state examination (MMSE) score changed by 1.226 ± 0.498 (from a baseline value of 19.45) in the control group and 0.9355 ± 0.5004 (from a baseline value of 17.42) in the sovateltide group from their respective baseline value. Neuropsychiatric Inventory (NPI) score was changed by −0.548 ± 1.123 (from a baseline of 9.516) in the control group, whereas by −1.968 ± 1.077 (from a baseline of 9.871) in the sovateltide group. These data indicate a potential beneficial effect of sovateltide in AD patients with late early to moderate stage of the disease.
5. Phase II—NCT04054414	Spinal Cord Injury (SCI)	Prospective, multicentric, randomized, double-blind, parallel, saline controlled to compare the safety and efficacy of sovateltide therapy along with standard supportive care in patients of acute spinal cord injury. Total spinal cord injury patients—40. Dose—0.3 μg/kg body wt. per dose. Three doses per day (total dose/day: 0.9 µg/kg body wt.) as an intravenous bolus over 1 min at every 3 ± 1 h on day 1, 3, and day 6. Primary objective—Testing safety and tolerability of sovateltide in spinal cord injury patients. Secondary objective—Assessment of International Standards for Neurological Classification of spinal cord injury (ISNCSCI), Walking Index for Spinal Cord Injury (WISCI) Score, Spinal Cord Independence Measure (SCIM) Score, Changes in Magnetic Resonance Imaging (MRI), Computerized Tomography (CT), and electromyography (EMG).	Expected to be completed in July 2022.

## Data Availability

Not applicable.

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
