# Peer review of "Sovateltide Mediated Endothelin B Receptors Agonism and Curbing Neurological Disorders"

_ijms, 2022, doi:10.3390/ijms23063146_

Round 1

Reviewer 1 Report

This manuscript summarises the drug development program for sovateltide, a selective agonist of the endothelin B receptor (ETBR), which appears to promote differentiation of neuronal progenitors to produce mature neuronal cells, and thus ameliorate effects of CNS damage.
The paper summarizes preclinical pharmacology and clinical safety and efficacy data for cerebral ischemic stroke, Alzheimer’s disease (AD), and acute spinal cord injury (SCI), and preclinical pharmacology data for efficacy in hypoxic-ischemic encephalopathy (HIE).

--------------
English quality is poor in places, e.g.
Line 142
"hypothesized ETARs"
to
"hypothesized that ETARs"

Line 164
"While deficiency"
to
"Deficiency"

Line 174
"ablated"
to
"abrogated"

Lines 291-292
"Drug-related adverse events were not occurred"
to
"No drug-related adverse events were reported" (if that is the case)

Line 311
"in" to "within"

Please check and polish.

-----------
Clinical trial information is out of date:

316-317 (stroke) "Our ongoing phase III trial is anticipated to be completed in April 2021."

452-453 (AD phase II) "The estimated  completion date of the study is August 2021."

566-567 (SCI phase I) "expected to be completed in June 2021".

Please update.

------------
There are adequate brief summaries of nonclinical pharmacology, but toxicology information is entirely missing; "After toxicological testing" (lines 230 and 435) is all that is said.

What were identified off-target receptors and toxicity target organs (and notably including developmental toxicity), at what doses, toxicokinetics, possible modes of action, etc...
This is a major deficiency from this reviewer's point of view.

-------------------

Reviewer 2 Report

The manuscript (review) by Ranjan and Gulati describes the role of endothelin-1 and endothelin receptors. It presents a novel drug – sovateltide, which acts as a selective agonist on endothelin B receptors. This novel drug can be used in treatment of stroke, AD, spinal cord injury, or neonatal hypoxic-ischemic encephalopathy. The manuscript mainly summarizes recent knowledge and results obtained from clinical trials performed by the company Pharmazz. The manuscript is interesting, attractive for the readers and fits with the journal (International Journal of Molecular Sciences). However, the quality is not enough to justify publication in the present form. I recommend major revision. Find below some specific suggestions to improve the quality of the paper.

Comments:

Keywords are missing in the abstract!

Chemical structure of the new drug should be added into introduction.

The references in the text should be in brackets!

The findings obtained from the clinical trials should be summarized in tables.

Page 2 – line 66 and 67 – “Approximately 800 GPCRs for endogenous ligands have been reported in human and mice” – please add an appropriate reference

Page 3 – line 114 and 115 – “All the clinical trials described in this review have been / being conducted / planned to be carried out are by Pharmazz Inc…….” – please rewrite this sentence or delete it

Page 4 – line 161 – abbreviations NP, GP should be explained.

Page 5 – line 200 – abbreviation DCX should be explained

Page 6 – line 242 – abbreviation ECG should be explained

Page 6- line 245 – abbreviation wt should be explained

Page 6 – line 272 and 273 – abbreviations NIHSS, mRS, BI should be explained

Page 6 – line 273 – abbreviations EQ-5D and SSQoL should be explained and the protocols should be cited

Page 7 – line 316 and 317 – “Our ongoing phase III trial is anticipated to be completed in April 2021” - please add actual information about the status of the phase III trial

Page 7 – line 322 – “According to a recent report……” – please add citation of the recent report at the end of the sentence

Page 7 and 8 – section Alzheimer´s disease – the immunotherapy of AD should be mentioned in the text

Page 8 – line 351 – abbreviation ECE-2 should be explained

Page 8 – line 354 – “….production of A 1-40 and A 1-42……” – I think it should be “…..production of Aβ 1-40 and Aβ 1-42……”

Page 9 – line 407 – abbreviations NGF and VEGF should be explained

Page 10 – line 452 and 453 – “The estimated completion of the study is August 2021” - please add actual information about the status of the phase II trial

Page 10 – line 458 – “…..and vis versa….” – is it correct???

Page 12 – line 566 and 567 – “…..and expected to be completed in June 2021.” - please add actual information about the status of the phase II trial

Page 13 – line 629 – abbreviation NICU should be explained

Page 13 – line 630 – abbreviation NIE should be explained

Page 14 – line 664 – the name of this section should be Conclusions

Round 2

Reviewer 1 Report

Thanks, typos corrected, and new section "Toxicological Studies" rudimentary but acceptable.

Reviewer 2 Report

The recommended template was not used.